# Pooled analysis of the association between mental health and violence against women: evidence from five settings in the Global South

Leane Ramsoomar [1,2] Andrew Gibbs [1,3] Esnat D Chirwa,[1,4]
Mercilene T Machisa [1,4] Deda Ogum Alangea,[5]
Adolphina Addoley Addo-Lartey,[5] Kristin Dunkle,[1] Rachel Jewkes [1,2]

¹Gender and Health Research Unit, South African Medical Research Council, Pretoria, Gauteng, South Africa
²School of Health Systems and Public Health, University of the Pretoria, Gauteng, South Africa
³Department of Psychology, University of Exeter, Exeter, Devon, UK
⁴School of Public Health, University of the Witwatersrand, Johannesburg, South Africa
⁵Department of Population, Family & Reproductive Health, University of Ghana School of Public Health, Legon, Accra, Ghana

**Correspondence to**
Dr Leane Ramsoomar;
leane.ramsoomar@mrc.ac.za

## ABSTRACT

**Objectives** To describe associations between men's poor mental health (depressive and post-traumatic stress symptomatology) and their perpetration of intimate partner violence (IPV) and non-partner sexual violence (NPSV), and women's mental health and their experiences of IPV and NPSV in five settings in the Global South.

**Design** A pooled analysis of data from baseline interviews with men and women participating in five violence against women and girls prevention intervention evaluations.

**Setting** Three sub-Saharan African countries (South Africa, Ghana and Rwanda), and one Middle Eastern country, the occupied Palestinian territories.

**Participants** 7021 men and 4525 women 18+ years old from a mix of self-selecting and randomly selected household surveys.

**Main outcome measures** All studies measured depression symptomatology using the Centre for Epidemiological Studies-Depression, and the Harvard Trauma Scale for post-traumatic stress disorder (PTSD) symptoms among men and women. IPV and NPSV were measured using items from modified WHO women's health and domestic violence and a UN multicountry study to assess perpetration among men, and experience among women.

**Findings** Overall men's poor mental health was associated with increased odds of perpetrating physical IPV and NPSV. Specifically, men who had more depressive symptoms had increased odds of reporting IPV (adjusted OR (aOR)=2.13; 95%CI 1.58 to 2.87) and NPSV (aOR=1.62; 95% CI 0.97 to 2.71) perpetration compared with those with fewer symptoms. Men reporting PTSD had higher odds of reporting IPV (aOR=1.87; 95% CI 1.44 to 2.43) and NPSV (aOR=2.13; 95% CI 1.49 to 3.05) perpetration compared with those without PTSD. Women who had experienced IPV (aOR=2.53; 95% CI 2.18 to 2.94) and NPSV (aOR=2.65; 95% CI 2.02 to 3.46) had increased odds of experiencing depressive symptoms compared with those who had not.

**Conclusions** Interventions aimed at preventing IPV and NPSV perpetration and experience must account for the mental health of men as a risk factor, and women's experience.

## STRENGTHS AND LIMITATIONS OF THIS STUDY

⇒ Synthesises data across Low to middle income country (LMIC) settings in the Global South using comparable measures.
⇒ Addresses the limited geographical scope of studies examining the association between perpetration of mental health violence against women and girls and experience in the Global South.
⇒ Addresses both intimate partner violence and non-partner sexual violence perpetration by men and experience by women using comparable measures.
⇒ Only two of the five studies are population-based, limiting generalisability.
⇒ All data are cross-sectional, limiting inference about causality and direction of effects.

## INTRODUCTION

Violence against women and girls (VAWG), particularly, intimate partner violence (IPV) and non-partner sexual violence (NPSV) are significant global public health problems. Globally, about 35% of women report having experienced either physical and/or sexual IPV or NPSV in their lifetime.[1] A large body of research has emerged identifying drivers of men's perpetration of physical and/or sexual violence against their female partners, and women's experience of this, in diverse contexts and populations. Drivers of violence perpetration and experience include poverty, gender inequalities, normalisation of violence, substance use, exposure to abuse as a child, and poor communication in relationships and conflict skills, as well as living in conflict-affected settings.[2–5] Research has established associations between experience of IPV and a range of negative health outcomes, particularly mental health outcomes, HIV, low birth weight, termination of pregnancy, suicidal ideation and harmful alcohol use.[1 6]

Poor mental health is increasingly recognised as both an outcome and a driver of VAWG.[7 8] Among women, the associations between VAWG and negative health outcomes have been recognised for some time and poor mental health has largely been established as an outcome of IPV and NPSV experience.[9] The WHO multi-country study on women's health and domestic violence found significant associations between lifetime experiences of partner violence and self-reported poor health, including more emotional distress, suicidal thoughts and attempts among abused, compared with non-abused, women.[2] A systematic review and meta-analysis of cohort studies found a strong association between recent IPV experience and depressive symptoms among women.[10] Research has shown that poor mental health may also be a risk factor for women's IPV experience. For instance, Devries and colleagues found that women who experience depression may be more accepting of partners with traits that predispose them to violence,[11] while other research found that women who experience depression may withdraw or display lethargy, which impacts their ability to seek help, or remove themselves from relationships.[10] Furthermore, the systematic review of cohort studies (mentioned above), found evidence of a bi-directional relationship between depression and IPV experience, whereby women's depressive symptoms were associated with subsequent IPV experience.[10]

Among men, research has found that poor mental health is associated with IPV perpetration, but the majority of this evidence has come largely from military or male offender samples from high-income countries.[12–14] There is, however, heterogeneity among samples and measures and definitions of IPV in previous work. Additionally this prior research has often focused on symptoms of post-traumatic stress disorder (PTSD),[8 13 15 16] with evidence indicating strong associations between PTSD and IPV perpetration.[13] Depression as a risk factor for male perpetration of violence against women has been less frequently studied and where examined, results have been mixed. For example, the cross-sectional population-based UN multi-country study in the Asia-Pacific region found that depression was associated with men's physical and sexual IPV perpetration in samples from four of six countries.[17] However, evidence from a systematic review of population-based studies found no association between depression and IPV perpetration.[18] This may be due to limited numbers of population-based studies on men's violence perpetration, and the frequent omission of measures of men's mental health in studies of IPV perpetration. To our knowledge the association between men's poor mental health and their risk of NPSV perpetration has not been examined in LMICs (low to middle income country/ies). However, IPV and NPSV are distinct types of violence, and where data are available, it is important to examine them both to investigate whether there are shared risk factors for men's perpetration of these and to understand the physical and mental health outcomes for women experiencing them.

Poor mental health often occurs against the backdrop of poverty, adverse childhood experiences, life traumatic events, low education, gender inequality, conflict and the wider context of generalised violence.[3 8 19 20] These factors operate either directly or indirectly to impact on childhood adverse events, poor mental health, substance use and relationship conflict, which in turn increase risk for IPV.[3] Data from a population-based survey in South Africa found that experiencing adverse childhood events (physical, sexual and emotional abuse) increased the risk of men's poor mental health (PTSD, depression), harmful alcohol use, which in turn increased the likelihood of male-perpetrated IPV.[8]

Although poor mental health has been recognised as both a driver and outcome of VAWG, the majority of studies are from high-income settings, focus largely on IPV only, and use differing measures of exposure and outcomes, which limits comparison across settings. Having comparable measures is foundational for making meaningful national, regional and global comparisons and understanding where prevalence and outcomes may differ.[21] In low-income and middle-income settings, where co-occurring public health challenges of VAWG, poor mental health and harmful alcohol use are prevalent, there is much less research on the associations between VAWG, poor mental health and VAWG. To strengthen the evidence base from low- middle income countries; advance the evidence base on the role of poor mental health in both IPV and NPSV perpetration and victimisation, we undertook a pooled analysis of baseline data. We used comparable measurement methods from five IPV prevention studies conducted in three countries in Africa (South Africa—which had two studies—Ghana and Rwanda) and one conflict-afflicted country in the Middle East, the occupied Palestinian Territories (oPt) (West Bank and Gaza). All studies report on recent (past 12 months) IPV and NPSV experience/perpetration, as opposed to lifetime exposure to IPV and NPSV. The paper aims to answer the following research questions: (1) Is men's poor mental health associated with IPV and NPSV perpetration ? (2) Is women's experience of IPV and NPSV associated with poor mental health outcomes?.

## METHODS

The studies included in this pooled analysis were conducted under the UK-Aid funded What Works to Prevent Violence Against Women and Girls? Global Programme (What Works). The primary goal of What Works was to advance the evidence base on the prevalence and drivers of VAWG in the Global South, and determine the effectiveness of interventions to prevent VAWG. The current study used the baseline data from 7021 men and 4525 women from five VAWG prevention studies in four countries (South Africa, Ghana, Rwanda and oPt (West Bank and Gaza)) to assess the association between poor mental health (depression and PTSD symptomatology) and IPV and NPSV perpetration among men

**Table 1** Data sets used for men and women included in the pooled analysis

| # | Study | Country | Study design | # clusters | N (men) | N (women) | Sampling or recruitment strategy | Age (years) |
|---|---|---|---|---|---|---|---|---|
| 1 | Evaluation of Stepping Stones and Creating Futures | South Africa | CRCT | 34 | 674 | 677 | Study volunteers | 18–35 |
| 2 | Evaluation of Sonke CHANGE Trial | South Africa | CRCT | 18 | 2406 | – | Household-based random sample survey | 18–45 |
| 3 | Evaluation of the RRS-COMBAT community intervention | Ghana | CRCT | 40 | 1973 | 1877 | Household-based random sample survey | 18+ (men) 18–45 (women) |
| 4 | Indashyikirwa -- couples intervention | Rwanda | CRCT | 28 | 1651 | 1660 | Volunteer recruitment from savings and loan association groups | 18–50 |
| 5 | Evaluation of Innovative Media to End VAWG through Community Education and Outreach in the oPt | oPt | Population-based survey | 55 | 308 | 371 | Population-based nationally representative survey | 18+ (men) 18+ (women) |

CRCT, cluster randomised control trial; oPt, occupied Palestinian territories; RRS, Rural Response System; VAWG, Violence against Women and Girls.

and the association between poor mental health and IPV and NPSV experience among women. These studies include the Stepping Stones and Creating Futures intervention (South Africa), the Sonke CHANGE Trial (South Africa), the Rural Response System (RRS (Rural Response System)-COMBAT (Community Based ActionTeams)) community intervention (Ghana), the Indashyikirwa couples intervention (Rwanda), and a population-based nationally representative survey in the oPt (West Bank and Gaza). Further information on the studies is available in the study sources referenced in table 1.

### Patient and public involvement

Patients and the public were not involved in the study design, implementation, or analysis presented in this paper. However, the individual projects had different levels of participant and public involvement as part of the wider What Works research uptake strategy.

### Measures

All study measures are presented in table 2. We assessed sociodemographics for men and women including age, current marital status, relationship residence status, education, employment in the past 3 months and past year, respectively. All studies except Indashyikirwa (couples) asked whether participants had worked in the past 3 months or the past year; the two South African studies asked whether female participants had worked in the past 3 months or not. The purpose of the past 3-month employment question was to give us an indication of recent employment. Income assessments in Indashyikirwa couples were not included in the present analyses, as they were tailored to the local context in which most participants engaged in subsistence agriculture.

Depression symptoms were measured using the previously validated Center for the Epidemiological Studies of Depression Short Form Scale.[22 23] PTSD symptoms were measured in three of the five studies (SSCF (stepping stones and creating futures), Sonke CHANGE Trial, Ghana (men only)), using the previously validated Harvard Trauma Questionnaire,[24] in settings where it was anticipated that the intervention would impact it. The Harvard Trauma Scale is a widely used cross-cultural measure to measure symptoms of post-traumatic stress,[24 25] and which has been used to measure PTSD symptoms in low-income to middle-income income settings.[8 26]

We measured men's current alcohol use using one item which asked men, 'Have you drunk alcohol in the past 12 months?'. Responses were either 'Yes' or 'No'. This is in keeping with international guidelines, which consistently measure current alcohol use as drinking at least one alcoholic drink in the 12 months preceding the baseline data collection,[27–29] while women (as a proxy) were asked if they had seen their partner drunk, and how frequently they saw them drunk in the past 12 months, as we did not always have access to the partner, and this was the most reliable measure of partner drunkenness in the past 12-month recall period. Violence perpetration (IPV and NPSV) and experience (IPV) were measured using items from the WHO Women's Health and Domestic Violence Survey,[30] modified appropriately to assess men's perpetration.[17] The NPSV Scale was first developed in South Africa[31] and subsequently refined and used extensively in the Asia-Pacific region.[32] There was no question on NPSV perpetration in the Indashyikirwa couples or the oPt studies, because of concerns about the particular sensitivity of the questions in those contexts.

**Table 2** Key measures

| Construct | Indicator | Definition |
|---|---|---|
| Violence against women | Physical IPV perpetration | Five items used to measure men's physical IPV perpetration in the last 12 months, for example, how many times (1) Did you slap your current or previous girlfriend or wife or throw something at her which could hurt her? (2) Have you pushed or shoved a current or previous girlfriend or wife? (3) Have you hit a current or previous girlfriend or wife with a fist or with something else which could hurt her? (4) Did you kick, drag, beat, choke or burn a previous or current girlfriend, partner or wife? (5) Did you threaten to use or actually use a gun, knife or other weapon against a previous or current girlfriend, partner or wife? Responses: 'Never', 'Once', 'A few times' or 'Many times'. Men who responded 'Once' or more to one or more items were coded as perpetrating physical IPV. The items were developed during the WHO Women's Health and Domestic Violence Survey, and UN Multi-country study on Men and Violence in Asia and the Pacific, modified to assess men's perpetration of physical violence and NPSV in the past year(17, 30). |
| | Physical IPV experience | Five items to measure women's violence experience in the past 12 months, for example, (1) How many times has a current or previous husband (or boyfriend) slapped you or thrown something at you which could hurt you? (2) Pushed or shoved you? (3) Hit you with a fist or with something else which could hurt you? (4) Kicked, dragged, beaten, choked or burnt you? (5) Threatened to use, or actually used, a gun, knife or other weapon against you?(30) Responses: 'Never', 'Once', 'A few times' or 'Many times'. Women who responded 'Once' or more to one or more items were coded as experiencing physical IPV. |
| | NPSV experience | Six items about NPSV perpetration in the past 12 months, for example, (1) How many times has any man who is NOT your boyfriend or husband forced or persuaded you to have sex against your will?... (2) Tried to force you to have sex against your will and did not succeed?... (3) Forced you to have sex against your will when you were too drunk or drugged to refuse? (4) Did two or more men force you to have sex with them at the same time against your will? (5) Did two or more men force you to have sex with them at the same time against your will when you were too drunk or drugged to refuse? (6) Was there an occasion when you agreed to have sex with one man and one or more others who you had not agreed to have sex with forced you to have sex with them as well? This was coded in the same way as for the men. |
| Alcohol use | Current alcohol use (past 12 months) | Current alcohol use by men was measured by asking one question about alcohol use in the 12 months preceding the baseline data collection: (1) Have you drunk alcohol in the past 12 months?. Responses were either 'Yes' or 'No'. |
| | Seen partner drunk | One item assessing if women has seen their partner drunk in the past 12 months. Responses: 'Yes' or 'No' |
| | Frequency of seeing partner drunk | One item assessing how often women had seen their partner drunk in the past 12 months. Responses were 'every day or nearly every day', 'Weekly', 'Once a month', 'Less than once a month', 'Never' and recoded into: 'never', 'occasionally' and 'frequently' |
| Mental health | Depression symptoms (men and women) | Three studies (SSCF, Sonke Change Trial and RRS-COMBAT) used the 20-item Centre for Epidemiological Studies Depression (CESD) Scale used to measure depression symptoms. Items were framed around statements about feelings in the past week, such as: 'During the past week I felt fearful' with responses, 'none or rarely', 'some or a little', 'moderate amount of time', and 'most or all the time'. Items were summed, and ranged between 0 and 41, with higher scores indicating more depressive symptoms (Cronbach's α ranged from 0.78 to 0.91 across studies). Three studies (Indashyikirwa and oPt) used the 10-item CESD Short Form Scale to measure depression symptoms. Items were summed, and ranged between 0 and 21, with higher scores indicating more depressive symptoms. |
| | PTSD symptoms (men and women) | Harvard Trauma Scale for PTSD, which reflected reactions to traumatic experiences in their lifetime experienced in the past week, such as: 'In the past week have you had recurrent thoughts or memories of the most hurtful or terrifying events' or 'In the past week have you had recurrent nightmares'. Responses were rated 0=not at all to 3=extremely. Items were summed with higher scores indicating more PTSD symptoms. Cronbach's α ranged from 0.91 to 0.94 across studies where PTSD was measured. |

IPV, intimate partner violence; NPSV, non-partner sexual violence; oPt, occupied Palestinian territories; PTSD, post-traumatic stress disorder ; RRS, Rural Response System; SSCF, stepping stones and creating futures.

## Data analysis

Descriptive statistics (frequencies and percentages) were used to summarise participants' sociodemographic characteristics within each study and in the pooled analysis. Within-study and pooled estimates considered any clustering with each study's sampling procedures. All pooled estimates were weighted according to the study sample size. We used forest plots, $I^2$ and Cochran's Q statistics to assess the consistency of outcomes across the studies. The $I^2$ values showed high heterogeneity in physical IPV as an outcome (80%, p<0.001 for men) and low heterogeneity for depression symptoms among women (14.8%, p=0.329). We used mixed-effects models to estimate overall effects and account for any heterogeneity across the studies due to methodological diversity. We fitted a one-stage individual patient data meta-analysis using mixed-effects logistic regression models to account for within-study and between-study variances (heterogeneity) across studies for both men and women.[17] We derived study-specific estimates and forest plots from a postestimation model of the mixed-effects logistic regression model. Both the main and postestimation models included participants' age as fixed effects. Prior to fitting models and in order to reduce any bias due to missing data, we examined patterns and levels of missing data in all key variables such as physical IPV, depression score, PTSD and NPSV. No missing data were encountered in the key variables for the women's data sets. In the men's data sets, we found no systematic patterns in the missing data and amount of missing data was minimal and ranged from 0.16% (Depression Score) to 1.3% (PTSD Score). We then used full information maximum likelihood to handle missing data in the mixed-effect models. We conducted sensitivity analysis to assess the impact of including the experience of childhood trauma on model estimates for studies that measured childhood trauma. We found non-significant change in model estimates. Thus, the final models were adjusted for participants' age and alcohol use or partner alcohol use (for women), because of the association between age and IPV experience/perpetration, and comorbidity between alcohol, VAWG and poor mental health found in previous research. All data were analysed using Stata statistical software V.17 (StataCorp, College Station, Texas, USA).

## RESULTS

### Men

The mean age of men across all studies was 33.1 years (SD=11.4). Of the men 45% were married while 45% were in a non-marital relationship. Overall, only 10% of men across all studies were not in a relationship; 61% of men indicated they were living with their partners, except in Rwanda (100%) and oPt (95%) where the majority were married. Almost a third of men (29%) reported not living together with their female partners. Most of the men had secondary school education or above (66%) and half (53%) the men in three studies were employed

in the past 3 months (we did not ask about employment status in Rwanda and the oPt) (table 3).

### Women

The mean age of women across the four studies was 31.3 years (SD=8.4). Over half (56%) of women were married, 38% were in a relationship, 75% were living with their partner or spouse, 19% were not living with their partner and 6% were not in a relationship. Nearly half (48%) of the women had completed secondary school education or above, and 53% reported working in the past 3 months, though this was not asked in Rwanda or the oPt. Just over a third of women (32%) reported that their partner consumed alcohol in the past month (see table 3).

### IPV and NPSV perpetration, poor mental health and alcohol consumption among men

Among the 7021 men across the studies, almost a third (29%) reported perpetrating IPV in the past year, with study-specific prevalence ranging from 12% (Ghana) to 50% (SSCF). Across the three studies where men were asked about perpetration of NPSV, 26% of men reported perpetrating NPSV in the past year, ranging from 10% in Ghana to 39% in SSCF. We did not ask about the perpetration of NPSV in Rwanda or oPt studies, as we were advised by the local partner that these questions would be too sensitive for these sociopolitical contexts. Overall, 29% of men reported depressive symptoms (range: 18%–46%), and in the three studies where we asked about PTSD symptomatology (SSCF, Sonke CHANGE Trial and Ghana), 6% reported PTSD (range: 5%–14.2%). In four studies where alcohol was asked about, 68% of men reported alcohol consumption in the past year (SSCF); 64% in the Sonke CHANGE Trial, 44% in RRS-COMBAT and 37% in Indashyikirwa. Alcohol consumption was not included in the questionnaire in the oPt (see table 4).

### Women

#### IPV and NPSV experience, poor mental health, and reports of partner alcohol consumption

Among the 4585 women across the studies, just over a third (31%) reported experiencing past year IPV, with study-specific prevalence ranging from 16% (Ghana) to 60% (SSCF). Across the two studies where NPSV was reported (SSCF and RRS-COMBAT), 11% of women reported experiencing past year NPSV, ranging from 3% in Ghana to 34% in SSCF. Overall, 36% of women reported depressive symptoms (range: 26%–45%). Only women in SSCF were asked about PTSD symptomatology, and 21% reported such symptoms (see table 4).

### Men

#### Association between poor mental health and men's violence perpetration

The association between men's depressive symptomatology and perpetration of past year IPV was significant across four studies, except for oPt, where the association was only suggestive (p=0.14) (table 5). The odds ranged from adjusted OR (aOR)=1.56 (95% CI 1.17 to 2.07) in

**Table 3** Sociodemographics, prevalence of alcohol consumption among men, reports of partner alcohol consumption among women in the included studies

| | Stepping Stones and Creating Futures Trial, South Africa | Sonke CHANGE Trial, South Africa | RRS-COMBAT, Ghana | Indashyikirwa *couples, Rwanda | oPt (Palestine) | All studies |
|---|---|---|---|---|---|---|
| | n (%)/mean (SD) | n (%)/mean (SD) | n (%)/mean (SD) | n (%)/mean (SD) | n (%)/mean(SD) | n (%)/mean (SD) |
| Men | (n=674) | (n=2406) | (n=1973) | (n=1651) | (n=317) | (n=7021) |
| Age | 23.8 (3.6) | 27.6 (5.7) | 39.3 (14.9) | 35.6 (7.1) | 43.9 (13.8) | 33.1 (11.4) |
| Current marital status | | | | | | |
| Married | 22 (3.3) | 448 (18.6) | 1271 (64.4) | 1095 (66.3) | 301 (95.0) | 3137 (44.7) |
| In a relationship | 508 (75.4) | 1539 (64.0) | 533 (27.0) | 556 (33.7) | 0 (0) | 3136 (44.7) |
| No relationship | 144 (21.4) | 391 (16.2) | 169 (8.6) | | 16 (5.0) | 720 (10.3) |
| Missing | 0 (0) | 28 (1.2) | 0 (0) | | | 28 (0.4) |
| Relationship residence status | | | | | | |
| Living together | 73 (10.8) | 932 (38.7) | 1314 (66.6) | 1651 (100) | 301 (95.0) | 4271 (60.8) |
| Not living together | 457 (67.8) | 1055 (43.9) | 490 (24.8) | | 0 (0) | 2002 (28.5) |
| No relationship | 144 (21.4) | 391 (16.2) | 169 (8.6) | | 16 (5.0) | 720 (10.3) |
| Missing | 0 (0) | 28 (1.2) | 0 (0) | | | 28 (0.4) |
| Education | | | | | | |
| None | 0 (0) | 0 (0) | 372 (18.9) | 265 (16.1) | 14 (4.4) | 651 (9.3) |
| Primary school | 77 (11.4) | 140 (5.8) | 334 (16.9) | 1088 (65.9) | 75 (23.7) | 1714 (24.4) |
| Sec school or above | 597 (88.6) | 2249 (93.5) | 1267 (64.2) | 298[18] | 227 (71.6) | 4638 (66.1) |
| Missing | 0 (0) | 17 (0.7) | 0 (0) | | 1 (0.3 | 18 (0.3) |
| Employed in the past 3 months | 240 (35.7) | 1192 (50) | 1231 (71.6) | n/m† | n/m | 2663 (52.7) |
| Employed in the past year | 156 (23.2) | 806 (33.7) | 1251 (63.4) | n/m | n/m | 2213 (43.8) |
| Drinks alcohol | 294 (43.6) | 948 (39.4) | 291 (15.0) | 246 (15.0) | n/m | 1779 (27.0) |
| Women | (n=677) | – | (n=1877) | (n=1660) | (n=371) | (n=4585) |
| Age in years, mean (SD) | 23.9 (3.6) | | 31.4 (8.5) | 32.7 (6.6) | 37.7 (11.7) | 31.3 (8.4) |
| Current marital status | | | | | | |
| Married | 29 (4.28) | – | 1068 (56.9) | 1096 (66) | 365 (98.4) | 2558 (55.8) |
| In a relationship | 524 (77.4) | – | 664 (35.4) | 564[34] | 0 (0) | 1752 (38.2) |
| No relationship | 124 (18.3) | – | 145 (7.7) | 0 (0) | 6 (1.6) | 275[6] |
| Relationship residence status | | | | | | |
| Living together | 113 (16.7) | – | 1303 (69.4) | 1660 (100) | 365 (98.4) | 3441 (75.1) |
| Not living together | 440 (65.0) | – | 429 (22.9) | 0 (0) | 0 (0) | 869[19] |
| No relationship | 124 (18.3) | – | 145 (7.7) | 0 (0) | 6 (1.6) | 275[6] |
| Education | | | | | | |
| None | 0 (0) | – | 401 (21.4) | 288 (17.4) | 3 (0.8) | 692 (15.1) |
| Primary school | 56 (8.3) | – | 426 (22.7) | 1115 (67.2) | 120 (32.4) | 1717 (37.5) |
| Sec school or above | 621 (91.7) | – | 1050 (55.9) | 257 (15.5) | 248 (66.8) | 2176 (47.5) |
| Employed in the past 3 months | 173 (25.6) | – | 1174 (62.7) | n/m | n/m | 1347 (52.8) |
| Employed in the past year | 97 (14.3) | – | 1034 (55.1) | n/m | n/m | 1131 (44.3) |
| Seen partner frequently drunk in the past year | 357 (52.7) | | 268 (14.3) | 742 (44.7) | n/m | 1367 (32.4) |
| Never | 320 (47.3) | – | 1609 (85.7) | 918 (55.3) | n/m | 2847 |
| Occasionally | 220 (32.5) | – | 115 (6.1) | 471 (28.4) | n/m | 806 |

Continued

**Table 3** Continued

| | Stepping Stones and Creating Futures Trial, South Africa | Sonke CHANGE Trial, South Africa | RRS-COMBAT, Ghana | Indashyikirwa *couples, Rwanda | oPt (Palestine) | All studies |
|---|---|---|---|---|---|---|
| Frequently | 137 (20.2) | – | 153 (8.2 | 271 (16.3) | n/m | 561 |

*Men only sample.
†Not measured.
oPt, occupied Palestinian territories; RRS, Rural Response System.

the RRS-COMBAT sample to aOR=3.45 (95% CI 2.64 to 4.52) in the Indashyikirwa sample. In the pooled analysis of all five studies, having depressive symptoms increased the odds for physical IPV perpetration among men aOR=2.13 (95% CI 1.58 to 2.87). In the three studies in which we asked about NPSV perpetration, the association between men's depressive symptomatology and NPSV perpetration was significant across the two South African studies (SSCF aOR=1.93 (95%CI 1.41 to 2.64) and the Sonke CHANGE Trial aOR=3.01 (95%CI 2.51 to 3.62)), but not the RRS-COMBAT Study in Ghana. In the pooled analysis (three studies), having depressive symptoms increased the odds for NPSV perpetration more than one half (aOR=1.62 (95%CI 0.97 to 2.71)), but this was not significant (p=0.06).

In the three studies where PTSD was measured, there were significant associations between men's PTSD symptomatology and past year physical IPV perpetration. Across individual studies (SSCF, Sonke CHANGE Trial and RRS-COMBAT) the adjusted association for

PTSD symptomatology and past year IPV perpetration ranged from aOR=1.79 (95%CI 1.15 to 2.79) in SSCF to aOR=2.51 (95%CI 1.74 to 3.62) in the Sonke CHANGE Trial (table 5). In the pooled analysis among men who reported PTSD, the odds of perpetrating past year physical IPV were increased almost twofold (aOR=1.87 (95%CI 1.44 to 2.43)), compared with those who did not report PTSD symptoms. PTSD symptomatology was also significantly associated with NPSV perpetration across all three studies, with adjusted associations ranging from aOR=1.70 (95% CI 1.10 to 2.62) in the Stepping Stones and Creating Futures Trial to aOR=3.08 (95% CI 2.13 to 4.45) in the Sonke CHANGE Trial. In the pooled analysis, the odds of perpetrating past year NPSV were increased more than twofold among men who reported PTSD (aOR=2.13 (95%CI 1.49 to 3.05)), compared with those who did not report PTSD symptoms.

Among women across all studies, experiencing IPV in the past 12 months was significantly associated with reported depressive symptomatology with adjusted associations

**Table 4** Prevalence of depression, PTSD symptomatology, physical IPV and NPSV perpetration among men and experience among women in the studies included

| | | Depression | PTSD | Physical IPV | NPSV |
|---|---|---|---|---|---|
| Men | | | | | |
| | N | n (%) | n (%) | n (%) | n (%) |
| SSCF-SA | 674 | 313 (46.4) | 96 (14.2) | 337 (50.0) | 261 (38.7) |
| Sonke-SA | 2406 | 700 (29.1) | 126 (5.0) | 952 (39.6) | 834 (34.7) |
| RRS-COMBAT-Ghana | 1973 | 619 (31.4) | 71 (3.60) | 235 (11.9) | 196 (9.9) |
| Indashyikirwa-Rwanda | 1651 | 290 (17.6) | n/m | 402 (24.4) | n/m* |
| oPt-Palestine | 308 | 112 (35.3) | n/m | 73 (23.0) | n/m |
| Overall | 7021 | 2034 (29.0) | 293 (5.8) | 1999 (28.5) | 1291 (25.5) |
| Women | | | | | |
| | N | Depression | PTSD | Physical IPV | NPSV |
| SSCF-SA | 677 | 306 (45.2) | 142 (21.0) | 403 (59.5) | 228 (33.7) |
| RRS-COMBAT-Ghana | 1877 | 695 (37.0) | *n/m | 290 (15.5) | 54 (2.88) |
| Indashyikirwa-Rwanda | 1660 | 429 (25.8) | n/m | 629 (37.9) | n/m |
| oPt-Palestine | 371 | 201 (54.2) | n/m | 94 (25.3) | n/m |
| Overall | 4585 | 1631 (35.6) | 142 (21.0) | 1416 (30.9) | 282 (11.0) |

*n/m, not measured.
IPV, intimate partner violence; NPSV, non-partner sexual violence; oPt, occupied Palestinian territories; PTSD, post-traumatic stress disorder ; RRS, Rural Response System; SA, South Africa.

**Table 5** Association between depression and PTSD symptoms and perpetration of physical IPV and NPSV

| | | Past year physical IPV perpetration | | | Past year NPSV perpetration | | |
|---|---|---|---|---|---|---|---|
| | All (N) | n (%) | aOR (95% CI)* | Study weight | n (%) | aOR (95% CI) | Study weight |
| **Depression** | | | | | | | |
| SSCF-SA | | | | | | | |
| No | 356 | 149 (41.8) | Ref | | 113 (31.7) | Ref | |
| Yes | 313 | 188 (60.1) | 2.14 (1.57 to 2.91) | 9.58 | 148 (47.3) | 1.93 (1.41 to 2.64) | 13.31 |
| Sonke-SA | | | | | | | |
| No | 1695 | 534 (31.5) | Ref | | 462 (27.3) | Ref | |
| Yes | 698 | 416 (59.6) | 3.22 (2.68 to 3.87) | 34.17 | 371 (53.0) | 3.01 (2.51 to 3.62) | 47.42 |
| RRS-COMBAT-Ghana | | | | | | | |
| No | 1354 | 145 (10.7) | Ref | | 139 (10.3) | Ref | |
| Yes | 619 | 90 (14.5) | 1.56 (1.17 to 2.07) | 28.27 | 57 (9.2) | 0.89 (0.64 to 1.23) | 39.26 |
| Indashyikirwa-Rwanda | | | | | | | |
| No | 1357 | 271 (20.0) | Ref | | n/m | n/m | n/m |
| Yes | 289 | 131 (45.3) | 3.45 (2.64 to 4.52) | 23.57 | n/m | n/m | n/m |
| oPt-Palestine | | | | | | | |
| No | 198 | 41 (20.7) | Ref | | n/m | n/m | n/m |
| Yes | 110 | 32 (29.1) | 1.46 (0.84 to 2.52) | 4.41 | n/m | n/m | n/m |
| Overall | | | | | | | |
| No | 4960 | 1140 (23.0) | Ref | | 714 (21.0) | Ref | |
| Yes | 2029 | 857 (42.2) | 2.13 (1.58 to 2.87) | 100 | 576 (35.3) | 1.62 (0.97 to 2.71) | 100 |
| **PTSD** | | | | | | | |
| SSCF-SA | | | | | | | |
| No | 572 | 276 (48.2) | Ref | | 212 (37.1) | Ref | |
| yes | 96 | 60 (62.5) | 1.79 (1.15 to 2.79) | 13.46 | 48 (50.0) | 1.70 (1.10 to 2.62) | 13.46 |
| Sonke-SA | | | | | | | |
| No | 2204 | 833 (37.8) | Ref | | 727 (33.0) | Ref | |
| Yes | 126 | 76 (60.3) | 2.51 (1.74 to 3.62) | 46.8 | 76 (60.3) | 3.08 (2.13 to 4.45) | 46.8 |
| RRS-COMBAT- Ghana | | | | | | | |
| No | 1902 | 220 (11.6) | Ref | | 180 (9.5) | Ref | |
| Yes | 71 | 15 (21.1) | 2.05 (1.14 to 3.68) | 39.75 | 16 (22.5) | 2.78 (1.56 to 4.96) | 39.75 |
| Overall | | | | | | | |
| No | 4678 | 1329 (28.4) | Ref | | 1119 (23.9) | Ref | |
| Yes | 293 | 151 (51.5) | 1.87 (1.44 to 2.43) | 100 | 140 (47.8) | 2.13 (1.49 to 3.05) | 100 |

*All models adjusted for participants age only.
aOR, adjusted OR; IPV, intimate partner violence; NPSV, non-partner sexual violence; oPt, occupied Palestinian territories; PTSD, post-traumatic stress disorder ; RRS, Rural Response System.

**Table 6** Association between IPV and NPSV experience and depressive symptoms among women

| | | Depression | | | |
|---|---|---|---|---|---|
| | | All (N) | n (%) | aOR (95% CI)* | Study weight |
| Physical IPV experience | | | | | |
| SSCF-SA | No | 274 | 94 (34.3) | Ref | |
| | yes | 403 | 212 (52.6) | 2.11 (1.54 to 2.90) | 14.77 |
| RRS-COMBAT | No | 1587 | 520 (32.8) | Ref | |
| | Yes | 290 | 175 (60.3) | 3.21 (2.47 to 4.15) | 40.94 |
| Indashyikirwa-Rwanda | No | 1031 | 193 (18.7) | Ref | |
| | Yes | 629 | 236 (37.5) | 2.69 (2.14 to 3.37) | 36.21 |
| oPt-Palestine | No | 277 | 134 (48.4) | Ref | |
| | Yes | 94 | 67 (71.3) | 2.83 (1.70 to 4.71) | 8.09 |
| Overall | No | 3169 | 941 (29.7) | Ref | |
| | Yes | 1416 | 690 (48.7) | 2.53 (2.18 to 2.94) | 100 |
| NPSV experience | | | | | |
| SSCF-SA | No | 449 | 167 (37.2) | Ref | |
| | Yes | 228 | 139 (61.0) | 2.64 (1.90 to 3.67) | 26.51 |
| RRS-COMBAT | No | 1823 | 663 (36.4) | Ref | |
| | Yes | 54 | 32 (59.3) | 2.82 (1.62 to 4.91) | 73.49 |
| Overall | No | 2272 | 830 (36.5) | Ref | |
| | Yes | 282 | 171 (60.6) | 2.65 (2.02 to 3.46) | 100 |

*All models adjusted for alcohol use and woman's age, except oPt, where model adjusted for age only.
aOR, adjusted OR; IPV, intimate partner violence; NPSV, non-partner sexual violence; oPt, occupied Palestinian territories; RRS, Rural Response System.

ranging from aOR=2.11 (95% CI 1.54 to 2.90) in SSCF to aOR=3.21 (95% CI 2.47 to 4.15) in RRS-COMBAT. In the pooled analysis, women who experienced IPV in the past 12 months, compared with those who had not, had a 2.5-fold increased odds of experiencing depressive symptoms than those who did not (aOR=2.53 (95% CI 2.18 to 2.94)).

Similarly, among women who had experienced NPSV in the past 12 months there were significant associations with depressive symptoms, with the adjusted associations ranging from aOR=2.64 (95% CI 1.90 to 3.67) in SSCF to aOR=2.82 (95% CI 1.62 to 4.91) in RRS-COMBAT. In the pooled analysis, women who experienced NPSV in the past 12 months had a more than 2.5-fold increased odds of experiencing depressive symptoms than those who did not (aOR=2.65 (95% CI 2.02 to 3.46)) (see table 6).

## DISCUSSION

VAWG and poor mental health are complex and overlapping problems. The results from this pooled analysis of data from men and women participating in five IPV prevention studies across Africa and in the Middle East showed evidence of a clear association between poor mental health and men's perpetration of both physical intimate violence and NPSV. Among women, we also found consistent associations between experiencing IPV and NPSV and reports of poor mental health outcomes.

There was a consistent positive association between men's depressive symptomatology and their perpetration of IPV in the pooled analysis, and in four out of the five country-level analyses. These findings are similar to those of the UN multicountry study which showed that men's depression was associated with physical and/or sexual IPV perpetration in three sites across Asia and the Pacific (Bangladesh, Cambodia and China),[17] and also reflects findings from studies in the Global North.[33] For example, in two meta-analyses,[34,35] it was found that depression was a moderate risk factor for male perpetrated violence against female partners.[34,35] In addition, a study in the USA found that men with depression showed an increased risk for perpetration of IPV.[36] There are a few potential reasons why men's depressive symptoms may be associated with their perpetration of IPV. It may be that men who are depressed engage in more drinking behaviour, which in turn increases their risk of IPV/NPSV perpetration. This is in keeping with other research that found poor mental health and alcohol are comorbid and this increased men's risk of violence perpetration,[18] as well as research from multiple low-income to middle-income settings which indicate that men's drinking places women at increased risk of IPV and NPSV perpetration.[37] It may also be that men who are depressed feel that they are unable to achieve traditional gender-role expectations placed on them, such as economic provision, or having

stable employment, and in turn, seek to exercise their gendered power by controlling and dominating their partners. Previous research in informal settlements found that among men in contexts of poverty, unemployment and social marginalisation, controlling their female partners has been used to consolidate hierarchy within social relationships and strengthen their self-evaluation of their performance as men.[38]

The reason for the lack of association between depressive symptoms and men's IPV perpetration in oPt is unclear. Previous work has found that exposure to political violence places men at risk for perpetration of IPV,[39] and this may mask other associations. Further work is required to understand the inter-relationships between exposure to political violence (not measured here), men's mental health and men's perpetration of IPV.

In this study, the association between men's depressive symptoms and perpetration of NPSV was significant in two of the three studies where NPSV was measured (SSCF and Sonke CHANGE Trials), but not in RRS-COMBAT; the latter finding may be due to small number of men reporting depressive symptoms and NPSV perpetration in the RRS-COMBAT intervention. Nonetheless, men who reported depressive symptoms did have an overall increased odds of perpetrating NPSV. An analysis of associations between past year NPSV perpetration and depression of men in a population-based sample from Bougainville, Papua New Guinea in contrast did not show associations.[40]

We found consistent positive associations between men's reports of PTSD symptoms and perpetration of IPV in four of the five studies and the pooled analysis, and for men's perpetration of NPSV, where this was significant in the pooled analysis and in all three of the individual studies (SSCF, Sonke and RRS-COMBAT) where NPSV was examined. There remains very little research on the association between PTSD and IPV and NPSV in general population samples. The evidence that there is primarily comes from largely military/offender populations,[12 13] and the analyses of these is complicated by the high rates of traumatic experiences that military staff may have experienced and other forms of violence in offender populations. The study of men in Bougainville, Papua New Guinea with very high population prevalence of PTSD and depression only found associations between past year depression and IPV perpetration, not with PTSD.[40] However, findings from our study extend the understanding of PTSD as a risk factor for IPV in general populations by showing that PTSD doubles the risk for NPSV perpetration. This advances the knowledge base, particularly in LMIC country settings, where NPSV, particularly in sub-Saharan African countries, is prevalent.

The associations between women's experience of IPV and NPSV and depressive and PTSD symptoms, reflects the broader literature base.[10] It also extends the knowledge base to include associations between NPSV experience among women and subsequent experience of PTSD. In addition, PTSD can be complex and often overlaps with depression among women. Therefore, disentangling the dimensions of PTSD and depression remains important to inform prevention programming and health service delivery among women. Prevention interventions need to address the mental health needs of women in prevention and response to IPV and NPSV.[41]

## STRENGTHS AND LIMITATIONS

This study's key strength is the pooled analysis of data from multiple settings and interventions across low-income to middle-income settings in the Global South, using comparable measures of men's IPV and NPSV perpetration, and women's experience of IPV and NPSV, and key covariates such as alcohol use. This addresses the limitation of many previous studies, which tend to focus on IPV only, and use differing measures of drivers and outcomes examining the associations between IPV and NPSV and mental health. The study also adds to the scant evidence base in the Global South, where the co-occurring problems of poor mental health and VAWG are prevalent. Furthermore, it highlights the complex and overlapping nature of alcohol, poor mental health as drivers and outcomes of VAWG, often against the backdrop of other structural drivers, such as poverty, food insecurity and unemployment. However, we also recognise limitations in the research. Previous research suggests that there is likely a bi-directional relationship between recent IPV and poor mental health, particularly for women.[10 42] However, the cross-sectional nature of the current data limits our ability to draw any conclusions about the temporal relationships, and necessitates future longitudinal analysis. Only two of the five studies were population-based and other studies were limited in generalisability, as they were based on populations recruited for the purpose of impact evaluation. Despite the current limitations, the study confirms that poor mental health is an important driver of men's use of violence, and women experiencing IPV and NPSV often experience poor mental health in LIMIC settings. We recognise that the study only has two measures of mental health symptoms. There may be previous psychiatric history accounting particularly for men's perpetration of both IPV and NPSV. However, we do not have data on a wider range of psychiatric symptomology from the current studies in these LMIC settings and, future studies examining risk factors for IPV and NPSV perpetration should take these into account.

## CONCLUSION

Our analysis has shown a consistent pattern of depression and PTSD symptoms as drivers of men's perpetration of IPV and NPSV, and increased depression and PTSD symptoms as outcomes among women who are in relationships characterised by IPV and NPSV. Interventions to address VAWG should more actively consider addressing poor mental health as driver, and address poor mental health

outcomes among women experiencing IPV and NPSV in LMIC settings in prevention and treatment programmes.

**Acknowledgements** The authors thank the research teams, programme staff and participants from all studies included in this pooled analysis.

**Contributors** LR conceptualised the analysis with AG and RJ. EC led the statistical analysis. AG was a PI on the Stepping Stones and Creating Futures study in South Africa, EC, DOA and AAL were co-PIs on the RRS-COMBAT Study in Ghana, KD was a PI on the Indashyikirwa Study in Rwanda, and MM is a Violence Against Women and Girls (VAWG) and mental health researcher, who contributed to the overall drafting of the manuscript. RJ was the Director of the overall What Works to Prevent VAWG Global programme and acts as guarantor. LR led the drafting of the manuscript; all authors contributed to comments and revisions.

**Funding** All studies presented and this pooled analysis of the data were funded through the What Works to Prevent Violence? A Global Programme on Violence Against Women and Girls (VAWG) funded by the UK Government's Department for International Development (DFID), Grant number: (PO 6254). However, the views expressed do not necessarily reflect the department's official policies and the funders had no role in study design; collection, management, analysis, and interpretation of data; writing of the report; and the decision to submit the paper for publication. Funding was managed by the South African Medical Research Council. Time drafting this manuscript was funded by the South African Medical Research Council.

**Competing interests** None declared.

**Patient and public involvement** Patients and/or the public were not involved in the design, or conduct, or reporting, or dissemination plans of this research.

**Patient consent for publication** Consent obtained directly from patient(s)

**Ethics approval** Ethical clearance for all studies was obtained prior to the studies commencing. For the Stepping Stones and Creating Futures Trial, clearance was obtained from the South African Medical Research Council's Ethics Committee (EC006-2/2015) and the University of KwaZulu-Natal's Biomedical Research Ethics Committee (BFC043/15). For the Sonke CHANGE Trial, ethical clearance was obtained from the University of Witwatersrand's Ethics Committee (M150443). For the Indashyikirwa couple's intervention in Rwanda, ethical approval was obtained from the Rwandan National Ethics Committee (340/RNEC/2015) and the South Africa Medical Research Council Ethics Committee (EC033-10/2015). A required research permit was obtained from the National Institute of Statistics Rwanda (0738/2015/10/NISR). For the RRS-COMBAT intervention in Ghana, the Noguchi Memorial Institute for Medical Research, University of Ghana ((# 006/15–16) and the South African Medical Research Council's Ethics Committee (EC031-9/2015) granted ethical clearance for the study. Ethical clearance for the oPt Study was granted by the South African Medical Research Council's ethics committee (EC014-5/2016). All studies followed ethical and safety guidelines for research on violence against women. All participants provided written informed consent before participation. Further information on the studies ethics is available in the study sources referenced in table 1.

**Provenance and peer review** Not commissioned; externally peer reviewed.

**Data availability statement** De-identified individual participant data for Stepping Stones and Creating Futures (South Africa), Sonke CHANGE Trial (South Africa), and Evaluation of the RRS-COMBAT intervention (Ghana) and oPt intervention, are available to anyone who wishes to access the data for any purpose at https://medat.samrc.ac.za/index.php/catalog/WW. De-identified individual participant data from the Indashyikirwa Couples Surveys (Rwanda) are available from the Principal Investigator of the study, Dr Kristin Dunkle: kristin.dunkle@mrc.ac.za, but may require permission from the Rwandan Ministry of Gender and Family Promotion (MIGEPROF) before transfer.

**ORCID iDs**
Leane Ramsoomar http://orcid.org/0000-0003-1934-579X
Andrew Gibbs http://orcid.org/0000-0003-2812-5377
Mercilene T Machisa http://orcid.org/0000-0001-7275-1100
Rachel Jewkes http://orcid.org/0000-0002-4330-6267

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
