## [Reviewer comments · BMJ Open]

ARTICLE DETAILS

TITLE (PROVISIONAL)	Pooled Analysis of the Association Between Mental Health and Violence Against Women: Evidence from five settings in the global South
AUTHORS	Ramsoomar, Leane; Gibbs, Andrew; Chirwa, Esnat; Machisa, Mercilene T.; Alangea, Deda; Addo-Lartey, Adolphina Addoley; Dunkle, Kristin; Jewkes, Rachel

VERSION 1 – REVIEW

REVIEWER	Mojahed, Amera Dresden University of Technology, Institute and Policlinic for Social and Occupational Medicine
REVIEW RETURNED	21-May-2022

GENERAL COMMENTS	I would like to thank the authors for the interesting manuscript. I believe it will bring much needed and valid attention to men's mental health as a driver of their perpetration of violence against women and girls. I however have few minor comments that I would like to see addressed, or at least argued within the manuscript. Minor comments: Page 5, lines 15-19: the authors wrote "To our knowledge the association between men's poor mental health and their risk of NPSV perpetration has not been examined in LMICs." Please elaborate more about non- partner sexual violence and why is it included here with physical IPV in the first place. Page 5, line 38: authors mentioned limited comparison when it comes to differing measures of exposure and outcomes. Why do we need comparison? Especially in the global south and when it comes to IPV or any other type of violence against women? Page 5, line 50: authors mentioned one conflict-afflicted country in the middle east, i.e., the occupied Palestinian Territories. Please consider using decolonial terms, i.e., Southwest Asian. Also, define/name these territories. Page 6, line 30: at the end of the first sentence, please specify "in this study" instead of "here". Page 8, line 8: What was the purpose of the past 3 months assessment point in some of the studies? Page 9: I recommend authors to add a column for points of assessment for all measures within the table. Page 10, line 15-20: Both measures for alcohol use (i.e., current use by men and seen partner drunk by women) are problematic.
--

	Please report how this measure was chosen/constructed to assess alcohol morbidity. In any way, an indication of its validity is needed. Page 15, line 12-13: Authors mentioned “We did not ask about the perpetration of NPSV in Rwanda or oPt as this was not the objective of these interventions.” Why not for these two countries/states? Page 9, line 37: the word “from” was doubled Page 21, line 10: either “use” or “seek”. Page 21, line 7-12: authors wrote about traditional societal norms and expectations of male gender roles and their role in men’s perpetration of VAWG. Then they moved to argue that female partners might have contributed to this perpetration by controlling their partners, who happen to fall under poverty, unemployment and social marginalisation (line 12-15). I recommend that the two points be clarified separately and with more nuance. Would female internalization of patriarchal gender norms contribute to their victimisation?
--	--

REVIEWER	Fazel, Seena University of Oxford, Psychiatry
REVIEW RETURNED	17-Sep-2022

GENERAL COMMENTS	This paper combines different LMIC samples to investigate associations between IPV and two psychiatric diagnostic categories (depression and PTSD). The outcomes are well defined. However, the measurement of the depression and PTSD is limited - diagnoses in these populations need more robust instruments (that incorporate more clinical judgement). The authors cite Cronbach alphas for the two instruments that are used - but this is a measure of internal consistency, but what is needed is clear evidence that the tools are concordant with a gold measure approach (such as a medical assessment informed by a clinical history). The Harvard tool is not well known - there are others with more evidence on their psychometric properties. As such, the study should be more careful in its language - and should focus on symptoms not diagnoses. The other main limitation is that the study does not adjust for many confounds - mostly age, and in table 6, for age and alcohol use (which is a crude categorical measure). There are many other confounds that could explain the association - from previous psychiatric history to socio-demographic variables (such as current levels of income or employment). Family psychiatric history will be a potential confound that is not considered. In addition, the study would benefit from examining a fuller range of psychiatric symptomology. This could provide some evidence of the internal validity of their approach (esp. if you see varying associations by diagnostic group), and it would allow for more clinical implications. These diagnoses are rarely without comorbidities in clinical practice - and depression, PTSD, and alcohol problems likely overlap. And in some people with personality problems. This could be examined in more detail. Finally, the paper did not discuss fully relevant evidence from high income countries - which could be compared with their findings.
---

VERSION 1 – AUTHOR RESPONSE

RE: ID bmjopen-2022-063730- Pooled Analysis of the Association Between Mental Health and Violence Against Women: Evidence from five settings in the global South		
Reviewer comments	Author changes	Position in the manuscript
*The following changes are in response to reviewer 1's comments		
1. Page 5, lines 15-19: the authors wrote "To our knowledge the association between men's poor mental health and their risk of NPSV perpetration has not been examined in LMICs." Please elaborate more about non-partner sexual violence and why is it included here with physical IPV in the first place.	Thank you for the comment. Non-partner sexual violence (NPSV) refers to sexual violence perpetrated, and experienced by someone who is not an intimate partner. There are two distinct and important types of violence (intimate partner violence and non-partner sexual violence) in the field of gender based-violence (GBV), and where data is available it is desirable to comment on both of them. For example, Breiding et al., 2017 studied IPV and sexual violence among 9,086 women and 7,421, other notable citations include (García-Moreno, et al. 2013 , Abrahams et. al, 2014 ; WHO, 2005). Importantly, it is critical to examine NPSV, given the extensive health consequences on women who experience it (unintended pregnancy , sexually transmitted infection (STI) and HIV, substance use and abuse, mental health issues). The decision to include NPSV is thus based on extensive previous literature in the field of GBV.	N/A
2. Page 5, line 38: authors mentioned limited comparison when it comes to differing measures of exposure and outcomes. Why do we need comparison? Especially in the global south and when it comes to IPV or	Thank you for the comment. The lack of comparability of between measures of exposures and outcomes has long been a criticism in the literature that measures GBV. Comparability is needed to ensure that we measure the same thing, are able to compare within and across regions and use this data to inform context specific prevention of, and response to GBV. A notable citation of the need for comparability come from, Ellsberg et al, 2015 , who argue that comparability is essential if researchers in the field are to reach consensus on methods that allow us to make meaningful comparisons across studies. In addition, the WHO multi country study on women's health and domestic	

any other type of violence against women?	violence against women found that it is very difficult to understand the similarities and differences in the prevalence, patterns and risk factors associated with violence in different settings if we do not have comparable measures.	
3. Page 5, line 50: authors mentioned one conflict-afflicted country in the middle east, i.e., the occupied Palestinian Territories. Please consider using decolonial terms, i.e., Southwest Asian. Also, define/name these territories.	Thank you for the comment. We have specified the names of the territories in the occupied Palestinian Territories. These are the West Bank and Gaza.	Page 4, line 28 Page 5, methods line 6 & 12
4. Page 6, line 30: at the end of the first sentence, please specify “in this study” instead of “here”.	Thank you for the comment. We have changed the sentence to read “ Patients and the public were not involved in the study design, implementation, or analysis presented in this study”.	Page 5, line 14-15
5. Page 8, line 8: What was the purpose of the past 3 months assessment point in some of the studies?	Thank you for the comment. As a point of clarity, we did not have a past 3 month assessment point for any exposure or outcome variables. We included only one demographic item on employment related to whether or not the participants were employed in the past 3 months. The purpose of the past 3 month employment question was to give us an indication of recent employment.	Page 7, line 5
6. Page 9: I recommend authors to add a column for points of assessment for all measures within the table.	Thank you for the comment. All analysis are based on baseline data only and we did not have 3 month assessment points for any exposure or outcome variables .	Abstract, line 5, Page 5, Methods, line 4-5

7. Page 10, line 15-20: Both measures for alcohol use (i.e., current use by men and seen partner drunk by women) are problematic. Please report how this measure was chosen/constructed to assess alcohol morbidity. In any way, an indication of its validity is needed.	Thank you for the comment. We measured current alcohol use by men using one item which asked men, “Have you drunk alcohol in the past 12 months?”. Responses were either “Yes” or “No”. This is in keeping with international guidelines, which consistently recommend measuring current alcohol use in this way (See, Tevik, 2021, WHO, 2000; NESARC, 2001; Parry, et al., 1998) We measured harmful alcohol use using the Alcohol Use Disorder Identification Test (AUDIT), a well-recognised scale (Saunders, et al. 1993) for detecting people with harmful alcohol consumption, and widely used to detect both harmful alcohol use generally and in the context of GBV (Babor et al., 1995 ; Chishinga et al, 2011) We used the measure of “Seeing partner drunk” and “How often did you see your partner drunk in the past year” as we did not have access to the partner. Therefore, asking women about their partner’s drunkenness and frequency thereof was the most reliable proxy, and in keeping with the past 12 month recall period of alcohol use measured in the study.	
8. Page 15, line 12-13: Authors mentioned “We did not ask about the perpetration of NPSV in Rwanda or oPt as this was not the objective of these interventions.” Why not for these two countries/states ?	Thank you for the comment. In the OpT (West Bank and Gaza), we did not ask about non-partner sexual violence, because we were advised by the local partner that these questions would be too sensitive for the socio-political context of the OpT. Similarly, Rwanda is not very democratic context, and we were advised that asking about NPSV might impact the of validity of reports, and the safety of participants. Hence we took an ethical decision not to ask about NPSV.	Page 7, lines, 18-19 Page 15, line 5-6
9. Page 19, line 37: the word “from” was doubled Page 21, line 10: either “use” or “seek”.	Thank you for the comment. We have deleted the repetition of the word “from”	Page 20, Discussion, line 2

10. Page 21, line 7-12: authors wrote about traditional societal norms and expectations of male gender roles and their role in men's perpetration of VAWG. Then they moved to argue that female partners might have contributed to this perpetration by controlling their partners, who happen to fall under poverty, unemployment and social marginalisation (line 12-15). I recommend that the two points be clarified separately and with more nuance. Would female internalization of patriarchal gender norms contribute to their victimisation?	Thank you for the comment. We believe the statement may have been misunderstood, as it is intended to convey that men control their female partners as a means to exercise their gendered power by controlling and dominating their partners. We have clarified this by changing the statement to read more clearly.	Page 21, lines 8-13
*The following changes are in response to reviewer 2's comments		
1. This paper combines different LMIC samples to investigate associations between IPV and two psychiatric	Thank you for the comment. We would like to clarify that the sole and primary intention of the measurement of PTSD and Depression in this study is not for diagnostic purposes, but for the purpose of being used in surveys to describe the epidemiology of mental health problems among the study population. To this end, we have used two very	

diagnostic categories (depression and PTSD). The outcomes are well defined. However, the measurement of the depression and PTSD is limited - diagnoses in these populations need more robust instruments (that incorporate more clinical judgement). The authors cite Cronbach alphas for the two instruments that are used - but this is a measure of internal consistency, but what is needed is clear evidence that the tools are concordant with a gold measure approach (such as a medical assessment informed by a clinical history). The Harvard tool is not well known - there are others with more evidence on their psychometric properties. As such, the study should be more	well recognised measures of both depression and PTSD The Centre for Epidemiological Studies-Depression (CESD) is widely used to measure symptoms of depression (Radloff, 1977) which has been validated for use in LMIC contexts (Jewkes et al, 2006; Murray et al, 2020); and in the general population (Vilagut, et al., 2016). We have also summed (range between 0-41) and used a cut-off point indicating possible depression. The Harvard Trauma Scale is also a widely used cross cultural measure to measure symptoms of Post-traumatic Stress (Mollica, et al. 1992; Darzi, 2017), and which has been used to measure PTSD symptoms in low to middle income settings (Machisa, et al, 2016; Christofides et al, 2018). We summed items with higher scores indicating more PTSD symptoms. Thank you for the comment. We agree with the recommendation to be more careful in the use of language relating to symptoms as opposed to diagnoses and have made changes throughout the manuscript to reflect this.	
---	---	--

careful in its language - and should focus on symptoms not diagnoses.		
2. The other main limitation is that the study does not adjust for many confounds - mostly age, and in table 6, for age and alcohol use (which is a crude categorical measure). There are many other confounds that could explain the association - from previous psychiatric history to socio-demographic variables (such as current levels of income or employment). Family psychiatric history will be a potential confound that is not considered.	Thank you for the comment. In determining what to adjust for, we chose variables that are known to be associated with both outcomes (IPV and MH) in the field of GBV. To our knowledge, family psychiatric history is not a known risk factor for IPV, and was neither measured in any of the current studies we analysed, nor adjusted for. We conducted a sensitivity analysis to assess the impact of adjusting for childhood trauma in the datasets where it was measured and found minimal change in the effect sizes (less than 2%).	
3. In addition, the study would benefit from examining a fuller range of psychiatric symptomology. This could provide some evidence of the internal validity of their approach (esp. if you see varying	Thank you for the comment. We agree that examining a fuller range of psychiatric symptomology would have benefitted the study. However, we do not have data on a wider range of psychiatric symptomology from the current studies, but will consider those known to be risk factors for IPV and NPSV in future work.	

associations by diagnostic group), and it would allow for more clinical implications. These diagnoses are rarely without comorbidities in clinical practice - and depression, PTSD, and alcohol problems likely overlap. And in some people with personality problems. This could be examined in more detail.		
4. Finally, the paper did not discuss fully relevant evidence from high income countries - which could be compared with their findings.	Thank you, we have included evidence from high income countries in the discussion.	Page, 20, lines 10-14

1

VERSION 2 – REVIEW

REVIEWER	Mojahed, Amera Dresden University of Technology, Institute and Policlinic for Social and Occupational Medicine
REVIEW RETURNED	07-Nov-2022
GENERAL COMMENTS	I thank the authors for the much enhanced MS. I have only one comment about the alcohol use measure: please cite references mentioned in your response.

VERSION 2 – AUTHOR RESPONSE

RE: ID bmjopen-2022-063730.R1- Pooled Analysis of the Association Between Mental Health and Violence Against Women: Evidence from five settings in the global South		
Reviewer comments	Author changes	Position in the manuscript
*The following changes are in response to the Editor		
**Previous comments now addressed with an explanation and in text references, as recommended by the editor		
1. Please revise the ‘Strengths and limitations of this study’ section of your manuscript (after the abstract). This section should contain up to five short bullet points, no longer than one sentence each, that relate specifically to the methods. The novelty, aims, results or expected impact of the study should not be summarised here.	Thank you for your comment. We have revised the ‘Strengths and limitations of this study’ section by making it more succinct, within the five bullet point limit and, related specifically to the methods. This is reflected in-text as follows:  • Synthesises data across LMIC settings in the global South using comparable measures. • Addresses the limited geographical scope of studies examining the association between mental health VAWG perpetration and experience in the global South • Addresses both IPV and NPSV perpetration by men and experience by women using comparable measures. Limitations  • Only two of the five studies are population-based, limiting generalizability. • All data are cross-sectional, limiting inference about causality and direction of effects. 	Page 3, Lines 3-10

2. Along with your revised manuscript, please include a copy of the STROBE checklist indicating the page/line numbers of your manuscript where the relevant information can be found (https://strobe-statement.org/index.php?id=strobe-home).	Thank you, we have included a copy of the STROBE checklist indicating the page and line numbers of our manuscript where the relevant information can be found	STROBE checklist uploaded
3. Please ensure that all reviewer comments are reflected by adequate modification to the text, not just explained in the point by point response. In particular, please ensure that the manuscript is updated to reflect your previous response to reviewer 2. For example, the manuscript should include justification for the variables chosen for the analysis, and any sensitivity analyses should be discussed.	Thank you. We have reflected modification to the text to reflect both reviewers previous and latest comments. For example, immediately below, we have responded to reviewer 1's, comments, and where changes were made in text we have indicated this, and its position in the manuscript.	Throughout the manuscript
The following changes are in response reviewer 1's previous comments, as addressed with modifications to the text)		
1. Page 5, lines 15-19: the authors wrote "To our knowledge the association between men's poor mental health and their risk of NPSV perpetration has not been examined in LMICs." Please elaborate more about non- partner sexual violence and why is it included here with physical IPV in the first place.	Thank you for your comment. Non-partner sexual violence (NPSV) refers to sexual violence perpetrated, and experienced by someone who is not an intimate partner. There are two distinct and important types of violence (intimate partner violence and non-partner sexual violence) in the field of gender based-violence (GBV), and where data is available it is desirable to comment on both of them. (García-Moreno, et al. 2013, Abrahams et. al, 2014; WHO, 2005). Importantly, it is critical to examine NPSV, given the extensive health consequences on women who experience it (unintended pregnancy, sexually transmitted infection (STI) and HIV, substance use and abuse, mental health issues). The decision to include NPSV is thus based on extensive previous literature in the field of GBV.	Page 5, Lines-10-13

	This is reflected in-text: “However, IPV and NPSV are distinct types of violence, and where data are available, it is important to examine them both to investigate whether there are shared risk factors for men’s perpetration of these and to understand the physical and mental health outcomes for women experiencing them .”	
2. Page 5, line 38: authors mentioned limited comparison when it comes to differing measures of exposure and outcomes. Why do we need comparison? Especially in the global south and when it comes to IPV or any other type of violence against women?	Thank you for the comment. The lack of comparability of between measures of exposures and outcomes has long been a criticism in the literature that measures GBV. Comparability is needed to ensure that we measure the same thing, are able to compare within and across regions and use this data to inform context specific prevention of, and response to GBV. A notable citation of the need for comparability come from, Ellsberg et al, 2015 , who argue that comparability is essential if researchers in the field are to reach consensus on methods that allow us to make meaningful comparisons across studies. In addition, the WHO multi country study on women's health and domestic violence against women found that it is very difficult to understand the similarities and differences in the prevalence, patterns and risk factors associated with violence in different settings if we do not have comparable measures. We have addressed in text and below “Having comparable measures is foundational for making meaningful national, regional and global	Page 5, lines, 24-26

	comparisons and understanding where prevalence and outcomes may differ .”	
3. Page 6, line 30: at the end of the first sentence, please specify “in this study” instead of “here”.	Thank you for the comment. We have changed the sentence to read “ Patients and the public were not involved in the study design, implementation, or analysis presented in this paper ”.	Page 6, line 25
4. What was the purpose of the past 3 months assessment point in some of the studies?	Thank you for the comment. As a point of clarity, we did not have a past 3 month assessment point for any exposure or outcome variables. We included only one demographic item on employment related to whether or not the participants were employed in the past 3 months. This is reflected in-text: “The purpose of the past 3 month employment question was to give us an indication of recent employment”.	Page 8, lines, 6-7
5. Page 15, line 12-13: Authors mentioned “We did not ask about the perpetration of NPSV in Rwanda or oPt as this was not the objective of these interventions.” Why not for these two countries/states?	Thank you for the comment. We have reflected this change is in-text: “There was no question on NPSV perpetration in the Indashyikirwa couples or the oPt studies, because of concerns about the particular sensitivity of the questions in those contexts” In the OpT (West Bank and Gaza), we did not ask about non-partner sexual violence, because we were advised by the local partner that these questions would be too sensitive for the socio-political context of the OpT.	Page 8, lines, 26-28 Page 15, line 6-7

	Similarly, Rwanda is not very democratic context, and we were advised that asking about NPSV might impact the validity of reports, and the safety of participants. Hence we took an ethical decision not to ask about NPSV.	
6. Page 21, line 7-12: authors wrote about traditional societal norms and expectations of male gender roles and their role in men's perpetration of VAWG. Then they moved to argue that female partners might have contributed to this perpetration by controlling their partners, who happen to fall under poverty, unemployment and social marginalisation (line 12-15). I recommend that the two points be clarified separately and with more nuance. Would female internalization of patriarchal gender norms contribute to their victimisation?	Thank you for the comment. We believe the statement may have been misunderstood, as it is intended to convey that men control their female partners as a means to exercise their gendered power by controlling and dominating their partners. We have clarified this by changing the statement to read more clearly in-text as follows: “It may also be that men who are depressed feel that they are unable to achieve traditional gender-role expectations placed on them, such as economic provision, or having stable employment, and in turn, seek to exercise their gendered power by controlling and dominating their partners. Previous research in informal settlements found that among men in contexts of poverty, unemployment and social marginalisation, controlling their female partners has been used to consolidate hierarchy within social relationships and strengthen their self-evaluation of their performance as men (36).”	Page 21, lines 7-13
The following change is in response to reviewer 1s latest comment)		
1. I thank the authors for the much enhanced MS. I have only one comment about the alcohol use measure Please cite references mentioned in your response.	Thank you for the comment. We have explained and cited the references used for the alcohol measures for (men's current use) and women's experience of partner drunkenness and frequency of drunkenness as follows	Page 8, Lines 17-23

	This is reflected intext: “We measured men’s current alcohol use using one item which asked men, “Have you drunk alcohol in the past 12 months?. Responses were either “Yes” or “No”.” This is in keeping with international guidelines, which consistently measure current alcohol use as drinking at least one alcoholic drink in the 12 months preceding the baseline data collection(25-27) , while women (as a proxy) were asked if they had seen their partner drunk, and how frequently they saw them drunk in the past 12 months, as we did not always have access to the partner, and this was the most reliable measure of partner drunkenness in the 12 month recall period.”	
*The following changes are in response to reviewer 2’s previous comments, addressed with modifications to the text, as recommended by the editor		
1. This paper combines different LMIC samples to investigate associations between IPV and two psychiatric diagnostic categories (depression and PTSD). The outcomes are well defined. However, the measurement of the depression and PTSD is limited - diagnoses in these populations need more robust instruments (that incorporate more clinical judgement). The authors cite Cronbach alphas for the two instruments that are used - but this is a measure of internal consistency, but what is needed is clear evidence that the tools are concordant with a gold measure approach (such as a medical assessment informed by a	Thank you for the comment. We would like to clarify that the sole and primary intention of the measurement of PTSD and Depression in this study is not for diagnostic purposes, but for the purpose of being used in surveys to describe the epidemiology of mental health problems among the study population. To this end, we have used two very well recognised measures of both depression and PTSD, the previously validated Center for the Epidemiological Studies of Depression Short Form (CESD-10) and the Harvard Trauma questionnaire We have indicated the use of both measures of depression and PTSD symptoms in cross cultural settings in text as follows:	Page 8, Lines-11-16

clinical history). The Harvard tool is not well known - there are others with more evidence on their psychometric properties. As such, the study should be more careful in its language - and should focus on symptoms not diagnoses.	“Depression symptoms were measured using the previously validated Center for the Epidemiological Studies of Depression Short Form (CESD-10) (22, 23). PTSD symptoms were measured in three of the five studies (SSCF, Sonke Change trial, Ghana (men only), using the previously validated Harvard Trauma questionnaire (24), in settings where it was anticipated that the intervention would impact it. The Harvard Tauma Scale is also a widely used cross cultural measure to measure symptoms of Post-traumatic Stress ((24, 25)), and which has been used to measure PTSD symptoms in low to middle income settings (8, 26).” Thank you for the comment. We agree with the recommendation to be more careful in the use of language relating to symptoms as opposed to diagnoses and have made changes throughout the manuscript to reflect this. We refer across the manuscript to depressive and post- traumatic stress symptomatology as opposed to Depression and PTSD diagnoses	Abstract, lines 3; 12 Tables 4-6 headings Page 6, methods, line 15 Page 15, results, lines 8, 20 Page 17, lines 3, 9, 16, 17, 21 Page 19, line 2 Page 20, Discussion, line 11
2. The other main limitation is that the study does not adjust for many confounds - mostly age, and in table 6, for age and alcohol use (which is a crude categorical measure). There are many other confounds that could explain the association - from previous psychiatric history to socio-demographic variables (such as current levels of income or employment). Family psychiatric history will be a potential confound that is not considered.	Thank you for the comment. In determining what to adjust for, we chose variables that are known to be associated with both outcomes (IPV and MH) in the field of GBV and those in LMIC settings. To our knowledge, family psychiatric history is not a known risk factor for IPV, and was neither measured in any of the current studies we analysed, nor adjusted for. We conducted a sensitivity analysis to assess the impact of adjusting for childhood trauma in the datasets where it was measured and, found minimal change	Page 11, Data analysis, lines - 20-25 Supplementary table 1

	in the effect sizes (less than 4%). We have indicated this in-text. We have also included a supplementary table indicating the adjusted ORs for models with childhood trauma and without childhood trauma. “We conducted sensitivity analysis to assess the impact of including the experience of childhood trauma on model estimates for studies that measured childhood trauma. We found non-significant change in model estimates. Thus, the final models were adjusted for participants’ age and alcohol use or partner alcohol use (for women), because of the association between age and IPV experience/perpetration, and co-morbidity between alcohol, VAWG and poor mental health found in previous research.”	
3. In addition, the study would benefit from examining a fuller range of psychiatric symptomology. This could provide some evidence of the internal validity of their approach (esp. if you see varying associations by diagnostic group), and it would allow for more clinical implications. These diagnoses are rarely without comorbidities in clinical practice - and depression, PTSD, and alcohol problems likely overlap. And in some people with personality problems. This could be examined in more detail.	Thank you for the comment. We agree that examining a fuller range of psychiatric symptomology would have benefitted the study. However, we do not have data on a wider range of psychiatric symptomology from the current studies, but will consider those known to be risk factors for IPV and NPSV in future work, particularly in LMIC settings. We have added this to the discussion This is reflected in-text as follows: “We recognise that the study only has two measures of mental health symptoms. There may be previous psychiatric history accounting particularly for men’s perpetration of both IPV and NPSV. However, we do not have data on a wider range of psychiatric symptomology from the current studies in these LMIC settings and, future studies examining risk factors	Page 23, Lines, 1-5

	for IPV and NPSV perpetration should take these into account.”	
4. Finally, the paper did not discuss fully relevant evidence from high income countries - which could be compared with their findings.	Thank you, we have included evidence from high income countries in the discussion. This is reflected in text as follows: “These findings are similar to those of the UN Multi-country study which showed that men’s depression was associated with physical and/or sexual IPV perpetration in three sites across Asia and the Pacific (Bangladesh, Cambodia, and China) (17), and also reflects findings from studies in the Global North(33). For example, in two meta-analyses, Stith et al. 2004 and Schumacher et al. 2001, found that depression was a moderate risk factor for male perpetrated violence against female partners(34, 35). In addition, a study in the USA found that men with depression showed an increased risk for perpetration of intimate partner violence(36).”	Pages, 20, lines 12-17 Page 21, lines 1-2